# Quantitative Analysis of Nuclear Lamins Imaged by Super-Resolution Light Microscopy

**DOI:** 10.3390/cells8040361

**Published:** 2019-04-18

**Authors:** Mark Kittisopikul, Laura Virtanen, Pekka Taimen, Robert D. Goldman

**Affiliations:** 1Department of Cell and Molecular Biology, Feinberg School of Medicine, Northwestern University, Chicago, IL 60611, USA; 2Department of Biophysics, UT Southwestern Medical Center, Dallas, TX 75390, USA; 3Institute of Biomedicine, Research Center for Cancer, Infections and Immunity, University of Turku, 20520 Turku, Finland; latevi@utu.fi (L.V.); pepeta@utu.fi (P.T.); 4Department of Pathology, Turku University Hospital, 20520 Turku, Finland

**Keywords:** lamins, structured illumination microscopy, single molecule localization microscopy, steerable filters, computational geometry, delaunay triangulation, voronoi tessellation

## Abstract

The nuclear lamina consists of a dense fibrous meshwork of nuclear lamins, Type V intermediate filaments, and is ~14 nm thick according to recent cryo-electron tomography studies. Recent advances in light microscopy have extended the resolution to a scale allowing for the fine structure of the lamina to be imaged in the context of the whole nucleus. We review quantitative approaches to analyze the imaging data of the nuclear lamina as acquired by structured illumination microscopy (SIM) and single molecule localization microscopy (SMLM), as well as the requisite cell preparation techniques. In particular, we discuss the application of steerable filters and graph-based methods to segment the structure of the four mammalian lamin isoforms (A, C, B1, and B2) and extract quantitative information.

## 1. Introduction

Nuclear lamins are type V intermediate filament proteins that assemble into dense fibrous meshworks composing the nuclear lamina (NL). The NL is located at the inner aspect of the nuclear envelope, where it provides a scaffold for organizing major nuclear structures, such as peripheral elements of heterochromatin and nuclear pore complexes [1,2,3]. Recent cryo-ET (Electron Tomography) studies have revealed that the NL is only ~14 nm thick [4,5]. This ultra-thin layer contains a complex of four densely-packed lamin isoform polymers: lamin A, C, B1, and B2 [1]. Therefore, defining the structure of the nuclear lamins within the NL in the context of the whole nucleus has required advances in super-resolution light microscopy and computational imaging techniques. Specifically, we have been able to take advantage of the advances in super-resolution light microscopy in combination with quantitative computer vision methods. With this approach, we have obtained detailed information on the localization and interactions of each of the lamins within the lamina. Indeed, the NL provides an opportunity to test the application of advanced imaging methods.

Furthermore, the detailed high-resolution architectures of the nuclear lamins are of significant biomedical interest due to their influence on the genome, as well as their involvement in a large number of different diseases. These laminopathies are caused mainly by mutations in the genes encoding the nuclear lamins, particularly in *LMNA*, the gene encoding lamins A and C. Over 600 different *LMNA* mutations cause a wide range of diseases, including Hutchinson-Gilford Progeria Syndrome, Emery-Dreifuss Muscular Dystrophy, dilated cardiomyopathy, and lipodystrophy [6,7]. Lamin B1 and B2 are two other isoforms encoded by *LMNB1* and *LMNB2*, respectively, and are less frequently associated with diseases. Duplication of the *LMNB1* gene has been associated with an autosomal dominant leukodystrophy in several families, and mutations in *LMNB2* have been linked to acquired partial lipodystrophy.

Prior to the last decade, researchers would obtain a series of images of the nuclear lamina acquired in distinct focal planes to form a Z-stack, typically using confocal immunofluorescence microscopy. The introduction of super-resolution microscopy has enabled the processing of multiple acquisitions to reconstruct an image beyond the Abbe diffraction limit of conventional light microscopy with fine optical sectioning. In the case of 3D-structured illumination microscopy (3D-SIM), phase-shifted sinusoidal illumination patterns are mathematically combined to obtain higher-resolution information [8]. For single molecule localization microscopy (SMLM)-based techniques, such as stochastic optical reconstruction microscopy (STORM), the location of single fluorophores are identified with a high level of precision by computational detection algorithms [8,9]. While both SIM and SMLM-based techniques allow for the examination of highly detailed structural information of the nuclear lamina, the analytical methods to extract information from the super-resolution data is an area of active research.

In this article, we will review the basic protocols for preparation of cells for super-resolution imaging of the nuclear lamina, as well as the computer vision-based image analysis techniques developed to gather quantitative data from those images.

## 2. Materials and Methods

### 2.1. Cell Preparation Methods

To prepare cells for microscopic examination of the lamins within the NL, we need to consider several factors for optimal imaging. Because the nucleus is typically not located near the surface of the cell attached to a glass coverslip, there are potential depth of field problems. Additionally, axial resolution in most light microscopy techniques is typically worse than the lateral resolution by a factor of two to three. This means that there are obvious advantages in using flatter cells to obtain larger in-focus areas of the nucleus. Further optimization is achieved by growing cells on glass coverslips for SIM imaging rather than glass slides.

Even though the lamina is a very thin layer, it is nonetheless a three-dimensional structure. Therefore, techniques need to be employed where volumetric imaging is possible and steps are taken to preserve the 3-D structure. We have found that methanol fixation of cells is particularly useful for single focal plane imaging of the NL because it is the ideal fixative for preserving intermediate filament proteins in general, and lamins in particular [10,11]. Moreover, this fixative dehydrates the cell and brings the nucleus closer to the coverglass. The small distance between the nucleus and the coverglass reduces issues with spherical aberration. In addition, to obtain optimal resolution we use Prolong Glass (ThermoFisher, Carlsbad, CA, USA) as a mounting medium, since this glycerol-based and hard-setting medium has a refractive index (RI) of 1.52, which closely mimics the RI of glass coverslips, and thereby provides optimal resolution due to the decrease in spherical aberration.

For this study, mouse embryonic fibroblasts (MEFs), courtesy of Yixian Zheng (Carnegie Institute), were seeded on round 12 mm coverslips (12-545-80, Thermo Fisher Scientific, Waltham, MA) and fixed and permeabilized with methanol for 10 min at −20 °C, followed by permeabilization with 0.1% Triton X-100 for 10 min at RT [10]. Cells were then stained for 1 h at RT with a rabbit polyclonal antibody to lamin A (1:500, Dechat et al.) in phosphate buffered saline (PBS, Gibco, Life Technologies, Carlsbad, CA, USA) containing 5% normal goat serum (Jackson ImmunoResearch Laboratories, West Grove, PA, USA), washed in PBS, followed by 1 h at RT with goat anti-mouse IgG-Alexa Fluor 488 (1:400, Life Technologies, Carlsbad, CA, USA). DNA was stained with Hoechst 33,258 for 1 h at RT (1:10000, Thermo Fisher Scientific). TetraSpeck beads (0.1µm, Life Technologies) were diluted 1:20 in the mounting medium. Finally, coverslips were mounted on slides in ProLong Glass (Life Technologies, Carlsbad, CA, USA).

Human dermal fibroblasts from a healthy donor (previously described in West, Gullmets et al., 2016 [12]) were seeded on glass bottom 35 mm dishes (MatTek, Ashland, MA, USA), fixed, and permeabilized with methanol for 10 min at −20 °C. Methanol fixation was used, since the flattening effect, as mentioned above, is helpful for Total Interference Reflection Fluorescence (TIRF), which is required for axial resolution with 2D-STORM. Cells were blocked for 30 min with 5% normal goat serum, stained for 1 h at RT with a rabbit polyclonal antibody to lamin A (1:500, 323; Dechat et al.), rinsed in phosphate buffered saline (Gibco, Life Technologies, Carlsbad, CA, USA), and then stained for 1 h at RT with goat anti-rabbit IgG-Alexa Fluor 647 (1:200, Life Technologies, Carlsbad, CA, USA). Cells on glass bottom dishes were kept in 1xPBS overnight at +4 °C and imaged the next day.

### 2.2. Imaging the Nuclear Lamins

#### 2.2.1. SIM

The 3D-SIM was carried out with a Nikon Structured Illumination Super-Resolution Microscope System (Nikon N-SIM, Nikon, Tokyo, Japan) using an oil immersion objective lens (CFI SR Apochromat 100x, 1.49 NA, Nikon). For 3D-SIM, 30 optical sections were taken at 60 nm intervals through the whole nucleus. Nikon Elements Advanced Research with an N-SIM module was used to reconstruct the structured illumination images. Illumination contrast, high-resolution noise suppression, and out-of-focus blur suppression were set with values 1, 0.75, and 0.15, respectively, for image reconstruction as in Table 1 below. For presentation, brightness and contrast were adjusted such that a majority of the data is visible in a linear range. Color shifts in the x-, y-, and z-axes were corrected using the TetraSpeck Fluorescent Microspheres (Life Technologies, Carlsbad, CA, USA).

#### 2.2.2. STORM

The 2D-STORM was carried out with a Nikon Stochastic Optical Reconstruction Microscopy (Nikon N-STORM, Nikon) using an oil immersion objective lens (CFI SR Apochromat 100x, 1.49 NA, Nikon). The focal plane was selected along with TIRF angle such that the region of the nuclear lamina closest to the adherent surface of the cell was visible. STORM acquires a time series of images and uses alternating illumination of fluorescent molecules to acquire and detect well-separated fluorophores [8]. Due to its photoswitching properties and high photon yield, Alexa Fluor 647 is a common dye used in STORM. The final image was reconstructed based on a total of 20,000 captured images using a capturing speed of 29 ms/frame. Nikon Elements Advanced Research with an N-STORM module was used to reconstruct the images. A density filter was used to reduce the background noise.

#### 2.2.3. SIM Image Reconstruction

SIM involves the acquisition of several images using a sinusoidal illumination pattern oriented at several angles and shifted by a set of equispaced phases to obtain super-resolution information. In 2D-SIM modes, this typically consists of three angles and three phases, whereas in 3D-SIM there are usually three angles and five phases. In order to process these for high resolution, images need to be reconstructed analytically in the Fourier domain after applying a Fourier Transform. Such reconstructions are typically achieved by the application of SIM reconstruction algorithms using software designed for each microscope system. However, open source and third-party alternatives are now available if needed.

The parameters used for SIM enhanced contrast eliminate high frequency noise, and for 3D-SIM, control the degree of optical sectioning. These parameters influence the reconstruction and modify the effective modulation transfer function (MTF) or related optical transfer function (OTF) through the application of filters in the Fourier domain. These filters may be distinct for each channel or wavelength collected. Other parameters include whether the illumination pattern angle should be detected from a single frame or if the highest quality angle should be taken from a set of frames, such as a z-stack. These parameters should be optimized for each wavelength and sample preparation condition, which is typically done by assessing image quality in both the Fourier and image domains. SIMCheck and other open source software have been made available to help assess the quality of reconstruction and to assess the presence of any artifacts.

For 3D-SIM there are two main ways to reconstruct an image. These include “stack” reconstruction and “slice” reconstruction. “Stack” reconstruction attempts to use data from multiple z-slices [13], whereas “slice” reconstruction only uses data from a single z-slice [14]. Within an experimental dataset, it is important to maintain a constant set of reconstruction parameters that do not vary among images. For our high-resolution studies of lamins we have used a Nikon SIM (N-SIM) microscope employing the slice reconstruction parameters for excitation at 488 nm for green fluorescent dyes and 561 nm for orange fluorescent dyes (Table 1). Depending on subject of interest, sample preparation, and imaging conditions, the Out-of-Focus Blur Suppression, a parameter only relevant to “slice” reconstruction, is the parameter we would first consider adapting if applied to another dataset.

### 2.3. Nyquist Sampling and SIM

To take advantage of the images acquired by SIM, the number of pixels must be increased to obtain the highest resolution. Resolution is defined as the smallest resolvable distance between two objects. Without super-resolution, the theoretical wavelength dependent resolution is
$$R_{x,y}=\frac{0.61\lambda }{NA}$$
according to the Rayleigh criterion, where λ is emission wavelength of light and *NA* is the numerical aperture of the objective. For emission of 525 nm and numerical aperture of 1.49, the resolution would be 215 nm. By utilizing the moiré effect, linear SIM increases the resolution by ~2X to ~115 nm [13]. Using the Nyquist-Shannon sampling theorem, this would necessitate a pixel size smaller than 57 nm. The Nikon N-SIM pixel size is usually calibrated to be around 32 nm, for example. Similarly, the axial resolution is calculated using the wavelength of the emitted light and the numerical aperture according to the Rayleigh criterion [15]:$$R_{z}=\frac{2\eta \lambda }{NA^{2}}$$

Here, the η is the index of refraction of the media. For NA = 1.49, λ = 525 nm, and η=1.52, $R_{z}=719\;nm$ for a widefield microscope. Three-dimensional structured illumination extends the axial resolution by ~2X to 360 nm. This suggests a z-step size of 120 to 180 nm for Nyquist sampling. Note that resolution calculated according to the Rayleigh criterion is usually more conservative than resolution derived from full width at half maximum (FWHM) measurements. The difference between the acquisition pixel size and resolution should be taken into consideration. The number of pixels is not directly indicative of spatial resolution, as long as the criteria of the Nyquist-Shannon sampling theorem are satisfied. Through interpolation and resampling, it is in fact possible to increase the number of pixels and decrease the pixel size without changing the spatial resolution. Oversampling, collecting pixels in excess of the requirements of Nyquist, is commonly used to improve the signal-to-noise ratio of the resulting image and decrease the error for the estimation for reconstruction parameters, but oversampling does not directly improve resolution [16].

### 2.4. Image Analysis

Images were primarily analyzed using MATLAB 2017a (Mathworks, Natick, MA, USA) with custom software. The steerableDetector MEX function for MATLAB was written by François Aguet (http://www.francoisaguet.net/software.html) as provided by the labs of Khuloud Jaqaman and Gaudenz Danuser in their common git repository [17]. The steerableDetector function was applied to reconstructed SIM images and a rastered STORM image with a fourth order filter. The Euclidean minimum spanning tree, Delaunay triangulation, and Voronoi tessellation of SMLM STORM data were computed using the Computational Geometry functions and the Bioinformatics Toolbox of MATLAB after loading the molecule list from Nikon Elements. The FIJI, Fiji is Just ImageJ, distribution with ImageJ 1.52i was used for initial evaluations of SIM data with the ImageJ plugin SIMCheck, as detailed in the text [18,19,20]. The Java Bio-Formats library (Open Microscopy Environment) was used to load the images in MATLAB and ImageJ.

### 2.5. Structure of the Nuclear Lamina

Cryo-electron tomography revealed that the nuclear lamina is composed of 3.5 nm lamin filaments [4,5]. While these structures are not directly resolvable by light microscopy, larger similarly oriented arrays of these filaments appear as fibers detectable by immunofluorescence with the super-resolution light microscopy techniques, including SIM and STORM. These fibers have been observed to form into a micrometer scale meshwork structure. When one of the lamin isoforms is depleted, the meshworks formed by the remaining lamin isoforms have variable density. Enlargement of faces and gaps in the meshwork formed by the remaining lamin isoforms have previously been quantitatively measured using the steerable filter technique reviewed here [10]. Knockouts of *Lmnb1*, *Lmna*, and *Lmnb2* in mouse embryonic fibroblasts (MEFs) all revealed meshwork face enlargement in decreasing order of prominence in our prior study. Along with the enlarged meshwork, large holes in the lamin A meshwork are particularly prominent in *Lmnb1*^−/−^ MEFs. For that reason, we chose a single *Lmnb1-null* MEF nucleus as a representative example for the majority of Figure 1, Figure 2, Figure 3, Figure 4 and Figure 5. While electron microscopy studies can examine the lamin meshwork at the scale of 1–10 nanometers, the field of view of the imaging is limited. The overarching goal of super-resolution light-microscopy studies and the computational image analysis has been to understand the structural organization of each of the fibrous meshworks assembled by the four lamin isoforms comprising the NL in the context of the entire nucleus.

### 2.6. Analysis of Structured Illumination Microscopy Images

The reconstruction of SIM images is conducted in the frequency domain of the image, as described above. Before applying detailed analysis of the fine structure in these images, it is critical to verify that the reconstructions are of high quality. The resulting images can be evaluated manually and with open source tools to assess the quality of the reconstruction.

Two initial manual qualitative assessments involve constructing the widefield image and assessing the two-dimensional Fourier transform of the reconstructed image. These are typically obtained by using the acquisition software provided by commercial vendors as part of the SIM reconstruction package, but can also be done by open source tools.

The widefield image is synthesized from the primary acquisition images by averaging the images across the distinct phases of each illumination angle (Figure 1A). The average across the phases for each angle should resemble a similar widefield image. The reconstruction of the widefield image is possible since the illumination intensity patterns are modeled as phase shifted sinusoidal patterns of the form $I\left(x,\Phi_{p}\right)=1+\cos\left(\nu x+\Phi_{p}\right)$, where x is the coordinate along the variation of the structured illumination pattern and $\Phi_{p}=\frac{2\pi \left(p-1\right)}{P}$ for *P* phase shifts, where usually P = 3 for 2-D SIM. Since the phase shifts are evenly distributed, averaging them results in a constant illumination pattern across the field of view for P > 1:$$\frac{1}{P}\sum_{p=1}^{p=P}\left[1+\cos\left(\nu x+\Phi_{p}\right)\right]=1$$

The principle is similar for 3D-SIM, where five phases are collected with variations over two dimensions. The synthesized widefield images for each angle should look nearly identical, since differences between them will violate the assumption that the same object is being imaged at each angle, which is required for accurate reconstruction.

The Fourier domain of the acquired (Figure 1B) and reconstructed images (Figure 1C,D) should also be assessed for their signal-to-noise ratio necessary for contrast and the presence of calibration features in the form of bright spots corresponding to the frequency of the illumination pattern. For visualization, the magnitudes of the Fourier coefficients are displayed using a log contrast lookup table. Equivalently, evaluating the log of the magnitude of the Fourier transform using a linear lookup table can be accomplished by calculating the log transform, according to the following equation as used in this manuscript:$$F_{log}\left(k_{x},k_{y}\right)=\log\left(\left|ℱ\left\{I\left(x,y\right)\right\}\left(k_{x},k_{y}\right)\right|+1\right)$$
where $F_{log}\left(k_{x},k_{y}\right)$ is the log-transformed magnitude of the Fourier transform, I(x,y) is either an individual acquired image at a particular phase and angle or a reconstructed image. From the Fourier domain, the resolution of the reconstructed image can be evaluated from a circularly averaged power spectral density.

Quantitative assessment of images acquired by SIM can be evaluated using open source tools. SIMCheck is available as a plugin for ImageJ [20]. SIMCheck provides a quantitative evaluation of the qualitative checks given above, as well as additional measures of reconstruction quality.

SIM images typically have a resolution improvement that is not fully isotropic due to the finite number of oriented illumination patterns used. Details are more resolvable along the angle of the illumination patterns used in SIM over the intervening angles, which is evident from the flower-like shape of the magnitude of the Fourier transform (Figure 1D). If the resolution improvement were isotropic, the Fourier transform would be circular. This has important implications for the analysis of reconstructed SIM images, as detailed in the next section.

To evaluate the resolution enhancement of the SIM reconstruction over the widefield image, a ratio can be taken of the SIM reconstructed image and either a widefield fluorescence image or a resolution reduced version of the reconstructed image (Figure 1E,F). This ratio will be used in later steps to help assess candidate features of the steerable filter, as described below.

### 2.7. Analysis of Lamin Fibers Using SIM and Steerable Filters

Steerable filters provide a way to detect curvilinear patterns within an image based on a particular scale and aspect ratio. These filters are used to detect the local orientation of structures in an image, which in turn is used to segment filaments.

Steerable filters are a type of linear filter, which in terms of image processing, process an input image and output a response image; where each pixel in the output is a linear combination of pixels in a neighborhood around the corresponding input pixel. Linear filters can be applied to an image using convolution in the spatial image domain or by pointwise multiplication in the Fourier domain due to the Fourier Convolution Theorem. A steerable filter can be represented either in the image domain (Figure 2E) or in the Fourier domain (Figure 2I). The image domain representation allows us to see the neighborhood around each pixel used by the filter via convolution (Figure 2E). In convolution, the filter is first centered at a pixel, multiplied pointwise across an area around the pixel, and then the products are summed together to find the response value at that pixel. This process is repeated for each pixel in the image resulting in the filter response image (Figure 2F). The Fourier domain representation makes the resolution and orientation selectivity of the filter apparent (Figure 2I). In the Fourier domain, the filter is applied by pointwise multiplication to the complex field of the image (Figure 2J). To obtain the filtered image from the pointwise product, the inverse two-dimensional Fourier transform is applied. The application of this filter to the reconstructed image (Figure 1C,D) is seen as a response image in the spatial (Figure 2F) and Fourier domains (Figure 2J).

Because both steerable filters and SIM reconstruction act upon the Fourier domain of images, the relationship of the orientation and resolution selectivity of the filter to the respective properties of the reconstruction process need to be considered. Since the resolution of a reconstructed SIM image is not isotropically uniform in the lateral dimensions (Figure 1D), orientation should not be assessed at the highest resolution available. Orientation measurements made at high resolution where information is not available equally in all directions would be biased towards the orientation of the structured illumination patterns used to acquire the images. Rather, spatial resolution should be selected at a scale where resolution is laterally isotropic. The spatial resolution selectivity of a steerable filter bank is assessed by calculating the mean filter across orientations, creating a Laplacian-like filter, which appears as a bright positive spot surrounded by a negative moat (Figure 2G). The scale selected by the steerable filters is proportional to the size of the positive spot. The Fourier domain representation of the mean filter typically resembles an annulus, since steerable filters are usually designed as bandpass filters (Figure 2K). Bandpass filters retain information around a certain spatial frequency, while attenuating or blocking smaller and larger frequencies (Figure 2L). When this mean filter is applied to the image, ridge features that match the scale selected have a higher response and are brighter (Figure 2H). While this would seem to negate the extended resolution of structured illumination, SIM does increase the magnitude of the Fourier coefficients within the low-resolution frequencies observable by widefield microscopy, and not just the higher frequencies contributing to extended resolution. Additionally, the original pixel intensities from the SIM reconstruction can be used to analyze the results of the steerable filter process, as outlined below.

Steerable filters can be used to either detect “ridges” (bright lines) or “edges” (lines of contrast between bright and dark areas). “Edge” detection by steerable filters are not considered further here, but may be useful in evaluating the width of lamin fibers in future studies. To clarify the terminology, the “ridges” detected here will be later referred to as edges of the lamin meshwork.

For the labeling of lamins by immunofluorescence for the detection of lamin fibers, ridge detection allows for the localization of the center of lamin fibers. This works in a two-step process by (1) determining the local orientation of a neighborhood around a pixel and (2) evaluating if that pixel has a value larger than its neighbors in the direction perpendicular to that orientation.

Steerable filters allow a response at any orientation to be interpolated from a finite set of oriented filter responses, as per the Nyquist-Shannon Sampling Theorem [21,22]. First, a set of steerable filters, called a filter bank, is used in combination to obtain a set of response images at distinct orientations (Figure 2E). A finite set of orientations are initially compared to an area centered at each pixel by convolving an oriented filter with the image as above. Effectively, this returns a score of how well the neighborhood around the pixel matches the oriented filter. This score is termed an orientation response for a particular angle (Figure 2B).

The orientation response can be interpolated from a finite sampling of orientations to form a continuous periodic function in terms of orientation angle. It is this property which makes these filters “steerable”, in that the filter orientation can be turned in any direction. From another perspective, a finite Fourier series in terms of orientation describes the periodic orientation response curves, meaning that the orientation response curves are band limited. The result is that steerable filters allow for the computation of a continuous response curve with respect to orientation. This property also means that steerable filters have limited orientation resolution, meaning that two orientations of equal intensity can only be distinguished if separated by a certain number of degrees.

Steerable filters first determine the best orientation for a neighborhood around each pixel (Figure 3A). The orientation angle assigned to a pixel is the angle where the orientation response function reaches an absolute maximum (Figure 3B). There are analytical and numerical methods for solving directly for the best orientation, so direct interpolation is not necessary. An orientation map can be generated by storing the best orientation angle as the value for each pixel (Figure 3A). The response for the filter oriented at that angle can also be stored for each pixel generating an orientation response map (Figure 2B and Figure 3A,B).

The center of a ridge is determined by scanning along a line in the space of the orientation response map perpendicular to the orientation to find the position where the orientation response reaches a maximum (Figure 3B). To encode this position in an image, a procedure called non-maximum suppression (NMS) is used. In NMS, pixels of the orientation response map that are not found to be the maximum in space along the perpendicular line are assigned a value of zero (Figure 3C). The resulting NMS image (Figure 3C), thus, contains non-zero pixels at the center of ridges in the image. The non-zero pixels have a value of the response corresponding to how well the neighborhood around them matches the filter steered to that orientation. The centers of these ridges then represent potential lamin fibers localized to the pixel grid.

To detect lamin fibers, the NMS image is processed by assessing the remaining non-zero pixels and the corresponding fluorescence values in the original image. The goal here is to compare the orientation response and fluorescence intensity values with the background values to separate true lamin fibers from false detections.

To evaluate the fluorescence intensity of immunostained lamins and the corresponding filtered response values, these intensities and their derivatives are compared to the statistical distribution of the background fluorescence intensity outside of the nucleus. While acquiring images, it is important to capture a sufficient amount of background for this purpose. To separate foreground, the nucleus, background, cytoplasm, and extracellular space, an initial rough segmentation of the nucleus is required.

The nucleus is initially segmented as a whole by thresholding the orientation response map (Figure 2B) and filling in enclosed areas of the binary outline. The objective here is to create a binary mask (Figure 2C) to identify the location of the nucleus by detecting an area enclosed by lamin fibers. For this enclosure to be detected, a continuous border of lamin fibers must be inferred. This initial threshold is determined by knowing the approximate size of the nucleus relative to the field of view of the image and then selecting a percentile from a distribution of orientation responses.

With a single threshold there may still be gaps in the detected structure. Some of these gaps are filled by including pixels near the already validated structure that are above a low threshold that is less selective than the original. In hysteresis thresholding, connected pixels that are higher than the low threshold are identified as connected components [23]. Only the connected components that also contain pixels that are above the high threshold are retained. Effectively, hysteresis thresholding identifies connected pixels likely to be lamin fibers that also contain pixels which are confidently lamin fibers.

The scheme we have employed is to select the 95th and 70th percentiles of the distribution of non-zero response values from the NMS map, and then use the two thresholds as high and low thresholds in the hysteresis thresholding process. The high threshold percentile should be adjusted to select pixels whose orientation response has a high confidence of selecting true lamin fibers. Here, we have heuristically assumed that the top 5% of steerable filter responses correspond to true detections of lamin fibers. The low threshold percentile should be selected by choosing an orientation response level that includes more lamin fibers but is less selective and may contain some false detections. Our selection assumes that the top 30% of steerable filter response intensities likely correspond to true detections. The idea is that for the purpose of identifying a rough mask of the nucleus, some false pixel detections may be acceptable as long as the nucleus is contained with the mask (Figure 2D). Morphological closing and opening are applied to close small gaps in the boundaries of the nucleus, and then the area enclosed by the white pixels in the processed NMS are filled in to produce a rough mask for the nucleus. This mask is then expanded slightly by morphological dilation (Figure 2C). The reconstructed image within the nuclear mask is shown in Figure 2D for comparison.

The coarse detection of the nucleus allows for determination of thresholds through the separation of foreground and background areas by statistical analysis. For example, the probability of a filter response within the nucleus being a true detection is evaluated by determining the probability that it belongs to a statistical distribution of responses outside of the nucleus (Figure 2C).

With the coarse mask, a threshold is determined by finding the response value that maximizes the difference between the statistical distributions of the response inside and outside the mask of the nucleus. This threshold is applied to produce a binary image (Figure 4D) from the NMS (Figure 4C), which still contains the non-suppressed values from the response (Figure 4A,B). Isolated segments containing less than five pixels are removed (Figure 4E), since the focus of this analysis is the meshwork structure, and short segments may correspond to false detections in the background.

A significant challenge occurs when lamin fibers intersect in an image. This may be because the lamin fibers are truly forming a junction or merely because they are crossing over or under each other in the z-dimension within the depth of focus of the 2D-image. For a steerable ridge filter, as outlined above, only a single orientation is detected, and thus only one lamin fiber is segmented at a time near an intersection (Figure 3B,C). Several algorithms are available to directly detect junctions with multiple orientations but are based on having prescribed templates.

One method to resolve the area around the junctions is to extend endpoints based on their orientation. In the case of the lamin meshwork, we extend lines according to the local orientation of the lines leading up to the endpoints until they become connected with another segmented pixel (Figure 4F). The extensions are then evaluated in a fashion similar to the calculation used to create the non-maximum suppression map (e.g., thresholding) during the later meshwork auditing process.

The next task is to analyze meshwork structure by classifying connected pixels as belonging to junctions (nodes) or edges in a graph theoretical sense. Again, to clarify, the graph edges discussed subsequently correspond with the computer vision “ridges” detected by the steerable filter. Junctions are initially identified by finding morphological branch points [24]. In a binarized skeleton made up of white pixels, morphological branchpoints are white pixels surrounded by three or more white pixels in their 8-connected neighborhood, where the intervening black pixels are not 4-connected (Figure 4G,H). Short segments of two pixels or less connecting the branchpoints are marked as being a part of the junction structure, as this indicates the convergence of many edges in close proximity. The pixels marked as being part of a junction are separated into connected components, and the location of the junction is then determined as being at the centroid of the connected components and reduced to a single pixel (white pixels, Figure 4J). Next, edges connected to the previous junction pixels are extended to those junctions (red pixels, Figure 4J). The endpoints of edges are either at a junction, where they connect to other edges, or they are freestanding and not connected to any other edges. Each edge is defined as an ordered set of pixels from one endpoint to the next. Pixels near junctions may be associated with more than one edge as the edges converge towards a junction. Once pixels have been assigned to edges and the location of junctions have been marked, the pixel-based representation of the meshwork is converted to an object-based representation, so that we can refine the meshwork further in terms of edges and junctions (Figure 4K). This means we will no longer be manipulating single pixels but rather we will be operating on entire edges at a time.

The object-based representation (Figure 5A) is further refined by removing edges based on the value of the pixels in either the reconstructed fluorescence image (Figure 5B), the ratio image between the reconstructed image and the low resolution image (Figure 5C), and the consistency of the fluorescence intensity along the edge (Figure 5D). The remaining edges with free standing endpoints are then also removed (Figure 5E). To perform these audits various thresholds are applied, including the mask-based threshold determined above. The other thresholds are automatically determined using Otsu’s method or Rosin’s method [25]. The exact procedure was detailed previously [10]. This results in the refined meshwork (Figure 5F,G). An overlay of the refined meshwork in magenta over the fluorescence in green is seen in Figure 5H, which is juxtaposed to the fluorescence image (Figure 5I) for comparison. The two-dimensional areas enclosed by the edges are regarded as faces.

Upon segmenting the lamin fibers, the distribution of junctions, edges which connect them, and faces surrounded by edges could be analyzed and quantified (Figure 5J). The quantitative evaluation of meshworks in this way could then provide insight into the effects of biological perturbations. For example, we previously used this to evaluate the effect of knocking out particular lamin subtypes and for characterizing individual lamin subtypes (Figure 5K–P).

The use of steerable filters in this fashion has limitations with regard to orientation and scale. In the scheme described above, only a single orientation is detected at each pixel and gaps are closed by extension of line segments. Junctions, however, could be detected through specialized filters or a method that can resolve and detect multiple discrete orientations. The scale of structures being imaged and analyzed may be variable when applied to other filamentous systems. In this case a steerable filter bank capable of evaluating multiple scales may be more appropriate than one optimized for a single scale.

### 2.8. Analysis of Single Molecule Localization Microscopy Images with Steerable Filters and Graph Theory

Single molecule localization microscopy (SMLM) techniques, such as STORM, do not produce an image directly, but rather result in a coordinate list of detected fluorophores generated from time-lapse series. Care must be taken to process the coordinate list to compensate for drift, thermal expansion, and multiple detections. To demonstrate the use of SMLM to study lamins, we have included a STORM image acquired using Total Internal Reflection Fluoresence (TIRF) microscopy of a human skin fibroblast labeled with Lamin A. A pixel-based image is made by accumulating the value of 2D Gaussian functions centered at each detected coordinate into a pixel grid (Figure 6A,B) and analyzing the resulting image as above (Figure 6C,D). This is equivalent to convolving the coordinate list with a 2D Gaussian. The width of the Gaussian determines the grid size by the Nyquist-Shannon sampling theorem. However, this approach is costly in time, and therefore a binning approach with a fine grid is often used for computational expediency [26]. For commercial microscopes, these operations are included in the vendor supplied software. ThunderSTORM, an open-source software package for ImageJ, also performs these processing and visualization steps [27].

SMLM detections can also be used to directly calculate a Fourier representation without the need for binning. Only the Fourier coefficients within the bandpass of the steerable filter bank (Figure 2K) would need to be calculated. The convolution method above provides a method for doing so but the choice of method is independent of the result. Steerable filters could then be directly applied to the Fourier representation, as above.

SMLM data of nuclear lamins has been collected by advanced microscopy techniques, but has not always been subjected to further analysis [28,29]. Where analysis has been done, graph theoretical methods on the coordinates directly have been used primarily for the purpose of visualization and limited quantification.

One class of graph-based methods operate by linking the localizations together via computational geometry (Figure 7). For example, a Delaunay triangulation links the localized fluorophores (Figure 7A) together to form triangles (Figure 7D, magenta). The triangles are chosen such that a circle drawn through the three localizations of the prospective triangle contains no other localizations. This is used to describe the space between fluorophores by examining either the edges of the triangles or the areas of the resulting triangles themselves [30]. Another method in this class is the Euclidean minimum spanning tree (EMST). The EMST contains the shortest total length of edges in the Delaunay triangulation required to connect fully all the localizations (Figure 7B, cyan). This method has been applied to segment the structure of nuclear lamins [9,31]. However, it is not clear how accurately the EMST as an abstract graph represents a physical structure and why this might be preferred over the Delaunay triangulation to describe lamin filaments.

A subgraph of the EMST is the nearest neighbor graph (NNG), where each localization is connected to the next nearest localization. Unlike the EMST, the NNG is not fully connected into a single graph, since the NNG only contains edges linking nearest neighbors and not the additional links needed to connect all the points into a single graph (Figure 7B, gold). The NNG is used to measure the nearest neighbor distances, and thus provide a measure of density and clustering. The nearest neighbor distances were used as a measure in a study that compensated for chromatic aberration by using activator-reporter dye pairs to show that A-type lamins formed denser meshworks than B-type lamins, and that lamin B1 was more proximal to the inner nuclear membrane than the A-type lamins [32].

The Voronoi tessellation falls in another class of graph-based methods, since this method partitions the space around localizations rather than between them [33,34,35]. The Voronoi diagram consists of edges that are equidistant between the two nearest fluorophore localizations (Figure 7C). The vertices of the localizations correspond to the centers of the circles drawn around the triangles of the Delaunay triangulation, making the Voronoi diagram the graph dual of Delaunay triangulation (Figure 7C,D).

A method relying on the fluorophore coordinate list directly has been developed to perform a similar operation to how steerable filters are processed [36]. The method proceeds by using an angular version of Ripley K’s function to determine orientation and traces fibers by assigning subsequent localizations to the fiber based on localized density and orientation [36]. The localized density is based on the area of the polygons contained within the Voronoi tessellation. This technique, however, has not been applied to the study of the nuclear lamina as of this publication.

Overall, both steerable filters and graph-based methods are used to segment and analyze SMLM microscopy data of filamentous structures, as formed by nuclear lamins. However, only the EMST technique has been directly applied to segment lamin fibers from SMLM data [9,31]. Steerable filters are applied to SMLM data either by creating an image via binning or by computing a Fourier transform as the initial step and proceeding as with a SIM image. Alternatively, the coordinates of individual fluorophore detections are processed directly through graph theoretical methods to evaluate clustering and density [32]. The advantages and disadvantages of these approaches are not clear, as a direct comparison of the methods and their utility in analysis has not been done.

## 3. Discussion

While super-resolution light microscopy has generated high-resolution images of the nuclear lamina, the analysis of those images requires further development in order to understand the structure of nuclear lamins, their relationship with each other, and their association with other molecular species in the nucleus. Here we have presented the application of steerable filters to structured illumination microscopy images and reviewed how steerable filters and related methods could be used to analyze single molecule localization microscopy data. We have also surveyed various graph-based methods for analyzing single molecule localization data, such as produced by STORM microscopy. Importantly, this review has further emphasized the need to develop these techniques for the specific microscopy method used.

In the examples provided, indirect immunofluorescence was applied for the convenience of using primary antibodies targeting the epitopes of interest, and secondary antibodies containing the fluorophores. Resolution can be further enhanced by either using direct immunofluorescence with fluorophores directly conjugated to the primary antibody, or by fusing the protein of interest with a fluorescent protein or tag [37,38,39,40]. This is especially true in the context of SMLM, where the effective resolution of 30 nm is of the same order of magnitude of the potential resolution enhancement ~30 nm of direct immunofluorescence or genetic tagging over indirect immunofluorescence. While direct immunofluorescence obviates the need for secondary antibodies, it loses the potential signal amplification associated with the secondary antibody technique. This caveat can be ameliorated with the advent of nucleic acid-based points accumulation for imaging in nanoscale topography (PAINT) labeling [41,42]. Nanobodies, such as single-domain camelid antibodies, also further enhance the attractiveness of direct immunofluorescence, but may be more difficult to manipulate and generate [43]. Genetic manipulation to add a fluorescence protein or a tag, may modify the expression, activity, and localization of the target protein structures of interest. While resolution can be further improved by simplifying the fluorescence labeling scheme, there are potential caveats that need to be considered in the experimental design.

New light microscopy techniques are constantly becoming available and are being applied to the study of nuclear lamins. For example, light-sheet microscopy techniques use orthogonal illumination to provide high resolution optical sectioning and reduced photobleaching, leading to images with isotropic resolution in three-dimensions [28,44]. Notably, these techniques can improve the axial resolution and depth of field over standard 3D-SIM, while providing more flexibility than the TIRF-STORM technique demonstrated here [45]. Further advances with light-sheet microscopy have integrated structured illumination or SMLM techniques to add super-resolution capabilities, allowing for the imaging of the entire nuclear lamina at high-resolution [44]. Nuclear lamins have already been used to demonstrate the power of these new techniques [28,44]. Furthermore, recent advances are making light-sheet microscopes simpler and more accessible to the biologist [46]. As light microscopy methods become increasingly sophisticated, the development of computational methods to analyze images of the nuclear lamina will have to keep pace.

## Figures and Tables

**Figure 1 cells-08-00361-f001:**
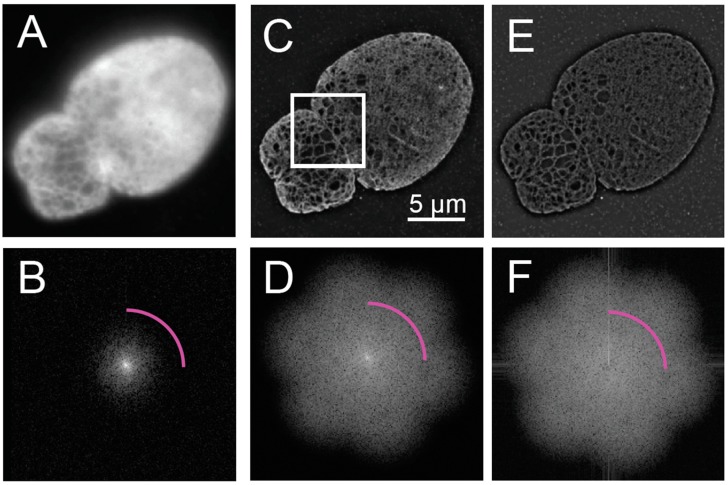
Structured Illumination Microscopy and Reconstruction of an Image of *Lmnb1*^−/−^ Mouse Embryonic Fibroblast. Fifteen frames, consisting of three angles and five orientations, for a 3D-Structured Illumination Microscopy (SIM) image were acquired of the nucleus of a *Lmnb1*^−/−^ mouse embryonic fibroblast (MEF) as part of a z-stack. The frames from a single z-slice were averaged to produce a widefield image (**A**) and the two-dimensional Fourier transform of the widefield image (**B**), showing the diffraction-limited resolution of widefield microscopy. The fifteen frames were processed into a reconstructed 3D-SIM image (**C**) and its two-dimensional Fourier transform (**D**). The ratio of (C) over (A) is computed by pixel-by-pixel division to show contrast enhancement of features in structured illumination microscopy (SIM) over widefield microscopy along with its two-dimensional Fourier Transform (**F**) of (**E**), showing retention of structural coefficients with reduced low frequency contribution. Magenta arcs in (B), (D), and (F) indicate resolution cut-off limit at 250 nanometers. Fourier Transforms are shown as a log (magnitude+1). The white scale bar indicates 5 micrometers. The white box in (C) is the area analyzed in Figure 4 A. The figure has been adapted from portions of Shimi and Kittisopikul et al.

**Figure 2 cells-08-00361-f002:**
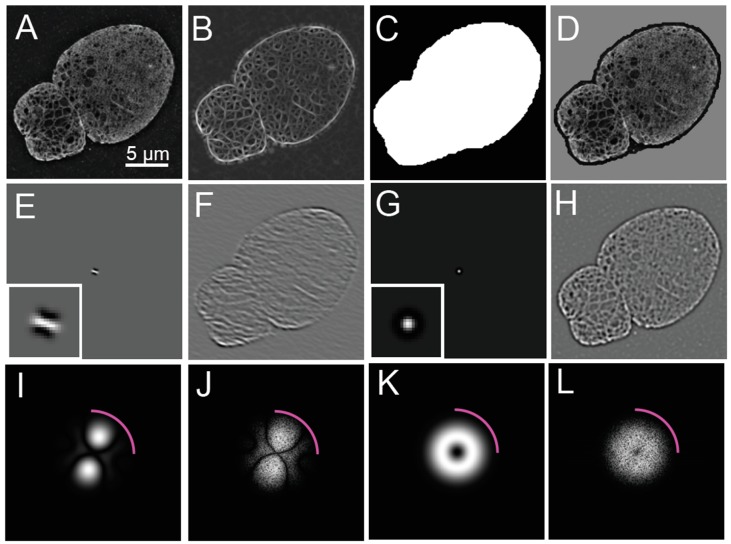
Steerable Filter Properties and Initial Application to a Reconstructed Image of a SIM Image of a *Lmnb1*^−/−^ MEF. (**A**) For reference, the reconstructed 3D-SIM image of a *Lmnb1*^−/−^ mouse embryonic fibroblast (MEF) nucleus is redisplayed from Figure 1C. (**B**) The image obtained after a steerable filter bank was applied to the image shown in (A). From the steerable filter response, a nucleus mask (**C**) was created by thresholding and morphological processing. To demonstrate that the nucleus was contained within the mask, the multiplicative product (**D**) of the reconstructed image (A) in the binary mask (C) is shown. To illustrate the process in more detail, a steerable filter orientated at 140 degrees (**E**) is shown in the image domain with an inset showing the filter magnified by five times. The grey background indicates zero-level, while black indicates the negative portion of the filter. When the steerable filter (E) was applied to the reconstructed image of the *Lmnb1*^−/−^ MEF nucleus (A), the response (**F**) indicates how well the image matches the oriented filter. Steerable filters were averaged over all orientation angles to produce the average filter (**G**) with the inset showing the averaged filter magnified by five times. The grey background indicates zero-level while black indicates the negative portion of the filter. The averaged filter (G) was applied to the reconstructed image (A) to produce an averaged steerable response (**H**), enhancing the areas matching the scale selective bandpass of the averaged filter. Two-dimensional Fourier transforms (2DFT) of (E), (F), (G), and (H) are shown in (**I**), (**J**), (**K**), and (**L**), respectively. The 2DFT of the oriented filter (I) and averaged filter (K) shows the scale and orientation selectivity in the Fourier domain. In the Fourier domain, the application of the oriented and averaged filters is a pointwise product of (I) and Figure 1D, resulting in (J) a pixel-wise product of (K), and Figure 1D, resulting in (L) due to the convolution theorem. Magenta arcs in (I-L) indicate widefield resolution cut-off limit shown at 250 nm as shown in in Figure 1. Fourier Transforms are shown as a log (magnitude+1). The white scale bar indicates 5 micrometers.

**Figure 3 cells-08-00361-f003:**
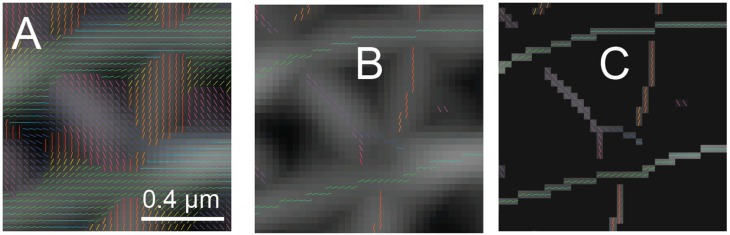
Orientation Detection and Non-Maximum Suppression of Steerable Filter Response of a SIM Image of a *Lmnb1*^−/−^ MEF. The orientation corresponding to the maximum response (argmax) of steerable filter bank (**A**) is overlaid on the maximum response value image as shown in Figure 2B. The orientation is indicated both by the slope of the lines drawn and their color according to an Hue-Saturation-Value (HSV) color map. Non-maximum suppression (NMS) is applied to the orientation indicators (**B**). The orientation indicators are only shown if the steerable filter response is larger than the interpolated values in either direction perpendicular to the orientation. The non-maximum suppression (NMS) is then applied to the maximum response (**C**) and orientation indicators (B). Response values are set to zero unless the steerable filter response is larger than the interpolated values in either direction perpendicular to the orientation. The white scale bar indicates 0.4 micrometers. The figure has been adapted from portions of Shimi and Kittisopikul et al.

**Figure 4 cells-08-00361-f004:**
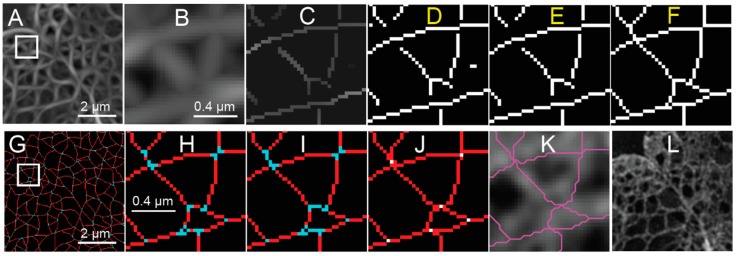
Pixel Based Processing of Steerable Filter Response and Non-Maximum Suppression to Form the Initial Meshwork Structure from a SIM Image of a *Lmnb1*^−/−^ MEF. The maximum steerable filter response (**A**) of the area indicated in Figure 1C of a reconstructed image of the *Lmnb1*^−/−^ MEF nucleus is shown. The area indicated by white box of (A) is magnified by five times (**B**). Non-maximum suppression (NMS) is then applied (**C**) to the maximum steerable filter response (A). The NMS (C) is then binarized such that non-zero pixels are indicated as white pixels to create a binarized image with line segments (**D**). Short segments of less than five pixels are removed from the binarized skeleton (**E**). The endpoints of line segments are extended until meeting another segment (**F**). Pixels in the extended binarized skeleton (F) for the area shown in Panel A are classified as either morphological edges (red) or branch points (cyan) (**G**). The area indicated by the white box of Panel G is magnified five times (**H**). Short segments between branch points are reclassified (F) as being part of a junction (cyan). The centroids of connected junctions (white) are identified (**J**) and the end points of edges are extended to them using the Bresenham line drawing algorithm (red). Edges are drawn as magenta lines and overlaid on reconstructed image of a reconstructed *Lmnb1*^−/−^ MEF (**K**). For comparison, the reconstructed fluorescence intensity image of the same area illustrated in (A) and (G) before processing is shown in (**L**). The white scale bar indicates two micrometers (A, G, L) or 0.4 micrometers (B–F, H–K). The figure has been adapted from portions of Shimi and Kittisopikul et al.

**Figure 5 cells-08-00361-f005:**
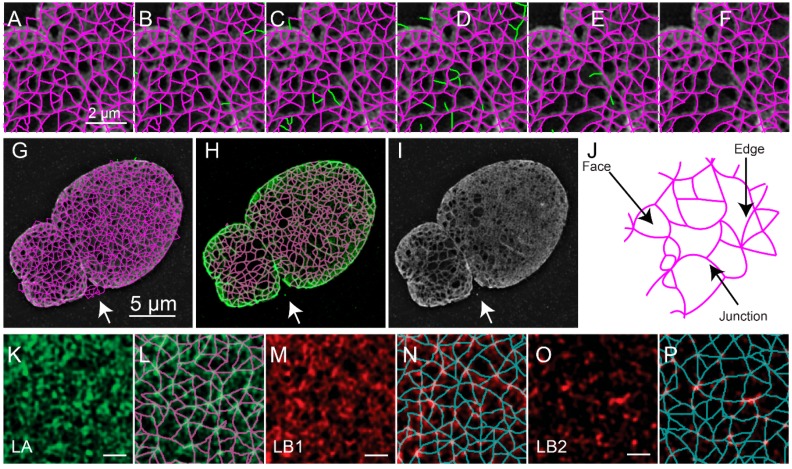
Edge and Face Based Auditing of Meshwork Structure of a SIM Image of a *Lmnb1*^−/−^ MEF. Unaudited edges are drawn as magenta lines over reconstructed image of the region in the white box indicated in Figure 1C (**A**). Edges are audited in a multistep process to remove edges which do not match features in the reconstructed image. First, edges (green) are removed due to the edges not matching the ratio image based on a distance weighted intensity measurement (**B**). Then, edges (green) are removed since either the minimum intensity, mean intensity, or normalized range of intensities in the reconstructed image (Figure 1C) do not meet thresholds described in the text (**C**). Next, edges (green) are removed since the edges do not sufficiently overlap the ratio image (Figure 1E) above an Otsu threshold (**D**). Finally, edges (green) are removed (**E**) since the new areas that result, faces as in (**J**) would better match the ratio image based on a distance weighted intensity measurement (**F**). These steps result in an audited meshwork shown in magenta (F). A zoomed-out view (**G**) of the entire unaudited meshwork is drawn as magenta lines over the reconstructed image. Green lines in (G) indicate edges not adjacent to faces being removed in early auditing steps. The audited meshwork drawn as magenta lines is superimposed on the reconstructed image intensity in green (**H**). The reconstructed image intensity as in Figure 1C is shown for comparison (**I**). The meshwork results in faces, edges, and junctions (J), whose properties have been quantified for further analysis [10]. Edges are lines tracing fibers of the nuclear lamina. Junctions are lines where multiple edges meet. Faces are enclosed areas surrounded by edges. The meshwork analysis has been applied to 3D-SIM images in dense Lamin A (**K,L**), Lamin B1 (**M,N**), and Lamin B2 (**O,P**) [10]. The white scale bar indicates two micrometers (A–F), five micrometers (G–I), or one micrometer (K–P). The figure has been adapted from portions of Shimi and Kittisopikul et al.

**Figure 6 cells-08-00361-f006:**
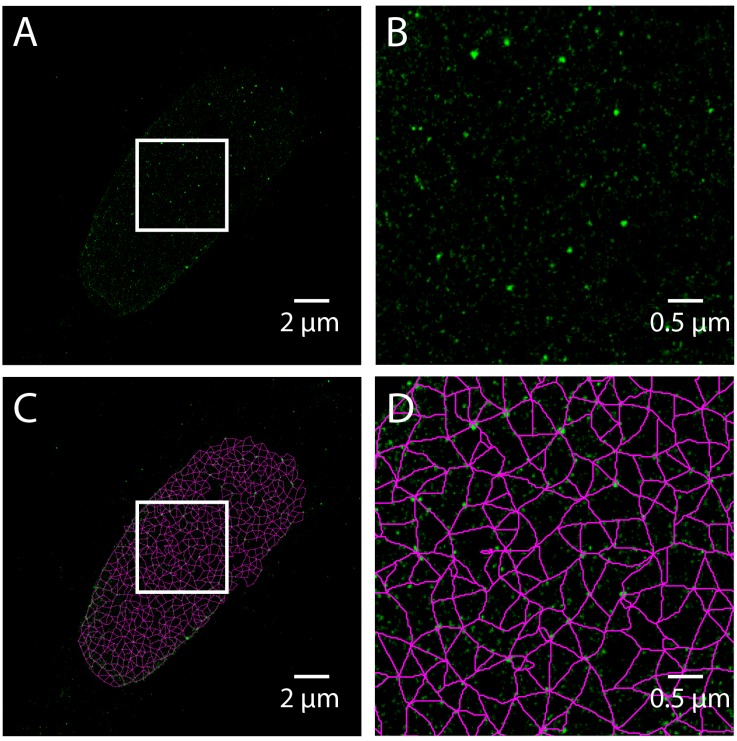
Steerable Filter Based Meshwork Analysis of a Total Internal Reflection Fluorescence – Stochastic Optical Reconstruction Microscopy (TIRF-STORM) Image of a normal Human Skin Fibroblast. TIRF-STORM image of Lamin A labeled human dermal fibroblast is rendered by binning fluorophore coordinates into a 20 nm pixel grid (**A**). The area in the white box (A) is magnified by four times (**B**). The steerable filter meshwork analysis as detailed for SIM applied (**C**) to the TIRF-STORM image (A). The area in the white box (C) is magnified by four times (**D**). The white scale bars are 2 micrometers (zoomed out) or 0.5 micrometers (zoomed in).

**Figure 7 cells-08-00361-f007:**
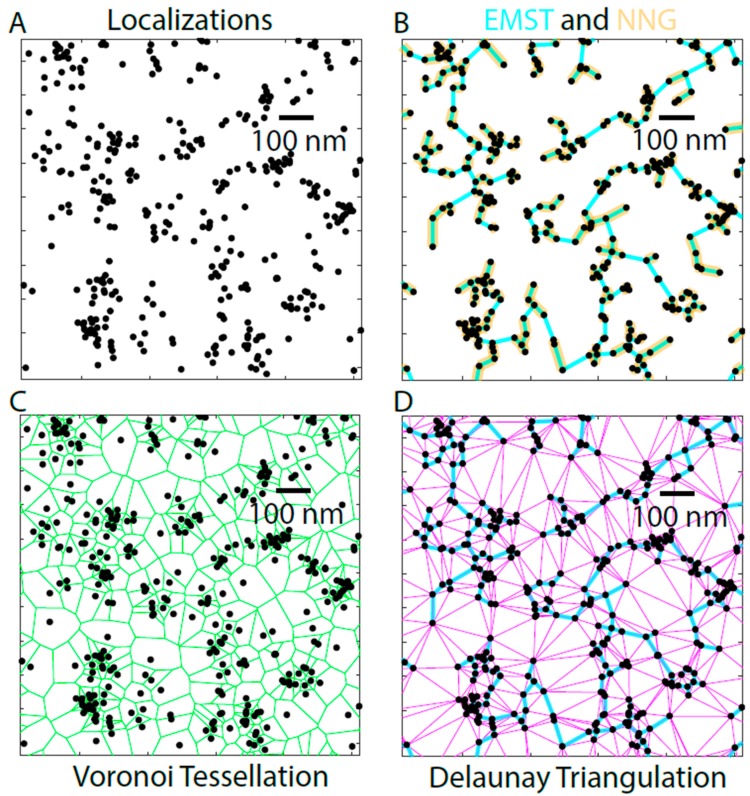
Graph Based Analysis of TIRF-STORM Image of a Human Dermal Fibroblast. TIRF-STORM localizations of fluorophores labeling Lamin A (**A**) are analyzed using several graph-based methods. The Euclidean Minimum Spanning Tree (EMST, cyan) connects all the detected fluorophores (dots) together using the shortest total line segment (**B**) and has been previously used to analyze lamin structures from single molecule localization microscopy [10]. A subgraph of the EMST is the Nearest Neighbor Graph (NNG, gold), which only connects fluorophore localizations with the next nearest localization without the requirement to connect all the fluorophores, as in EMST, and has been used to evaluate the relative density of A and B-type lamins [32]. The Voronoi Tessellation (**C**) draws lines (green) equidistant from two fluorophore coordinates resulting in polygons. The area within each polygon is the area closer to the single detected fluorophore contained within it than to fluorophores outside of the polygon and is used to approximate the reciprocal of the local fluorophore density. The Delaunay Triangulation [33,34] (**D**) (magenta) draws triangles between all the detected fluorophores, such that a circle drawn around the triangles does not contain another detected fluorophore, producing a more highly connected graph than the EMST. The EMST (cyan) is a subset of the edges of the Delaunay Triangulation (D) and the graphs are used to analyze the space between fluorophores. The nodes of the Voronoi are the centers of the circles going through the vertices of each triangle in the Delaunay Triangulation. Black scale bar is 100 nm.

**Table 1 cells-08-00361-t001:** N-SIM Reconstruction Parameters for Lamin Intermediate Filaments.

Parameter Name	Parameter Value
Illumination Contrast Modulation	1.00
High Frequency Noise Suppression	0.75
Out-of-Focus Blur Suppression	0.15

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
