# Peer review of "Quantitative Analysis of Nuclear Lamins Imaged by Super-Resolution Light Microscopy"

_cells, 2019, doi:10.3390/cells8040361_

Round 1

Reviewer 1 Report

With this manuscript Kittisopikul et al. provide an extensive review on data rendering of structured illumination microscopy (SIM) and single molecule localization microscopy image data sets of nuclear lamin assemblies. They discuss their own previous work in the context of the work of other groups working with similar techniques. With impressively clear figures they describe the rationale behind the application of steerable filters and graph-based methods and how quantitative information can be deduced from the calculated images. This review is certainly a very valuable resource for all researchers working on filamentous cellular structures with microscopic superresolution methods, as these methods are also applicable to microtubule and actin networks. I strongly recommend publication in “Cells” after a few minor corrections have been made.

p. 3, line 79: as most researchers still work with widefield microscopy and conventional confocal microscopy, I would suggest correct the word “half”, as axial resolution is only one third of the lateral resolution with these methods.

p. 3, line 92: it may make sense to add that flatter cells and a lower distance between nuclei and coverslip surfaces also reduce problems with spherical aberration.

p. 3, line 100 and Discussion: in the context of the methods and calculation methods described in this review it would make sense to discuss the effect of the size of a primary/secondary antibody couple (up to 30 nm) on resolution compared to the use of directly labeled nanobodies.

p. 3, line 107: the flattening effect was described above and the sentence is therefore redundant

p. 5, line160: it should be explained somewhere above that image processing requires a Fourier transformation.

p. 5, lines 164-171 are duplicated and should be deleted.

p. 11, line 367: I think it should read “...respective properties of the reconstruction process...”

p. 14, line 463: Is Figure 2A,C meant instead of B,C?

p. 20, line 661: this sentence is not clear to me.

p. 20, line 666: I think it should read “...required to fully connect...”

Generally I would recommend using a smaller character set for the figure legends in order to separate them more clearly from the main text.

Author Response

We thank the reviewer for the kind comments and the close reading of our manuscript. We agree that many of the techniques could be applied with little modification to filamentous structures such as microtubule and actin networks, especially when used with super-resolution microscopy techniques.

p. 3, line 79: as most researchers still work with widefield microscopy and conventional confocal microscopy, I would suggest correct the word “half”, as axial resolution is only one third of the lateral resolution with these methods.

We agree that this could use further clarification. We have changed the language to state "Additionally, axial resolution in most light microscopy techniques is typically worse than the lateral resolution by a factor of two to three", included an equation for the determining axial resolution according the Rayleigh criterion, and stated an example axial resolution calculation for widefield and SIM.

p. 3, line 92: it may make sense to add that flatter cells and a lower distance between nuclei and coverslip surfaces also reduce problems with spherical aberration.

We have added language to discuss how this lower distance reduces problems with spherical aberration and included how the refractive index matching of the mountant also addresses spherical aberration.

p. 3, line 100 and Discussion: in the context of the methods and calculation methods described in this review it would make sense to discuss the effect of the size of a primary/secondary antibody couple (up to 30 nm) on resolution compared to the use of directly labeled nanobodies.

We have added a paragraph to the discussion to the address the effect of labeling technique on resolution.

p. 3, line 107: the flattening effect was described above and the sentence is therefore redundant

We have altered the language to refer to the prior description and to emphasize the importance of flattening to TIRF.

p. 5, line160: it should be explained somewhere above that image processing requires a Fourier transformation.

We have added references to the Fourier domain in the SIM Image Reconstruction section.

p. 5, lines 164-171 are duplicated and should be deleted.

The duplicated lines have been removed.

p. 11, line 367: I think it should read “...respective properties of the reconstruction process...”

The correction has been made.

p. 14, line 463: Is Figure 2A,C meant instead of B,C?

We have split the figure reference to clarify that the "orientation response map" is Figure 2B and the binary mask is Figure 2C.

p. 20, line 661: this sentence is not clear to me.

The explanation of Delaunay triangulation has been clarified to read: "The triangles are chosen such that a circle drawn through the three localizations of the prospective triangle contains no other localizations. "

p. 20, line 666: I think it should read “...required to fully connect...”

The correction has been made and rearranged to avoid a split infinitive.

Generally I would recommend using a smaller character set for the figure legends in order to separate them more clearly from the main text.

We defer to the publisher on the formatting of figure legends. They were smaller on the initial submission.

Thank you again for the review. We appreciate the comments and feel the suggestions have strengthened the manuscript.

Reviewer 2 Report

All in all, I recommend the manuscript authored by Kittisopikul et al. to be published in Cells. I have only some minor changes to suggest.

Line 48: LMNA should be written in italic letters.

Line 67: Dash is missing: SMLM based

Line 105: Insert space after "healthy donor"

Line 106: Is Chromotek meant here?

Line 166: Space missing after z-slices

Line 171- 179: Repetition of lines 164-171

Line 219: Space missing after 3.5 

Line 223: Delete the second full stop after STORM

Line 228: Italicise Lmnb1, Lmna, Lmnb2.

Line 232: Italicise Lmnb1-/-, Lmnb1(-null)

Line 242: "to check" is a bit informal. Please rephrase.

Line 298: Delete space 3D- Structured Illumination Microscopy

Line 309: Size of the scale bar is already given in the figure panel and therefore unnecessary to be repeated in legend. Please check in all figure legends. Colour information is also unnecessary. 

Line 310: "The white box in (A) is..." I assume C is meant here.

Line 343: Insert space after mask

Line 615: Delete space after filters

Line 755: Should be cell

Author Response

We thank the reviewer for the close reading and comments.

Line 48: LMNA should be written in italic letters.

Italics letters have been used.

Line 67: Dash is missing: SMLM based

The dash has been added

Line 105: Insert space after "healthy donor"

The space has been added

Line 106: Is Chromotek meant here?

The company has been clarified to be MatTek.

Line 166: Space missing after z-slices

The space has been added

Line 171- 179: Repetition of lines 164-171

The repeated lines have been removed

Line 219: Space missing after 3.5 

The space has been added

Line 223: Delete the second full stop after STORM

The full stop has been removed

Line 228: Italicise Lmnb1, Lmna, Lmnb2.

The gene names have been italicized

Line 232: Italicise Lmnb1-/-, Lmnb1(-null)

The gene names has been italicized

Line 242: "to check" is a bit informal. Please rephrase.

We have made the language more formal: "Before applying detailed analysis of the fine structure in these images, it is critical to verify that the reconstructions are of high quality"

Line 298: Delete space 3D- Structured Illumination Microscopy

The space has been deleted.

Line 309: Size of the scale bar is already given in the figure panel and therefore unnecessary to be repeated in legend. Please check in all figure legends. Colour information is also unnecessary. 

We will confer with the editor regarding the preferred formatting of the figure legends and scale bars.

Line 310: "The white box in (A) is..." I assume C is meant here.

This has been correct.

Line 343: Insert space after mask

The space has been added

Line 615: Delete space after filters

The space has been deleted

Line 755: Should be cell

The word has been corrected.

We thank the reviewer for the comments and formatting advice.